# Radiotherapy Plan Quality Assurance in NRG Oncology Trials for Brain and Head/Neck Cancers: An AI-Enhanced Knowledge-Based Approach

**DOI:** 10.3390/cancers16112007

**Published:** 2024-05-25

**Authors:** Du Wang, Huaizhi Geng, Vinai Gondi, Nancy Y. Lee, Christina I. Tsien, Ping Xia, Thomas L. Chenevert, Jeff M. Michalski, Mark R. Gilbert, Quynh-Thu Le, Antonio M. Omuro, Kuo Men, Kenneth D. Aldape, Yue Cao, Ashok Srinivasan, Igor J. Barani, Sean Sachdev, Jiayi Huang, Serah Choi, Wenyin Shi, James D. Battiste, Zabi Wardak, Michael D. Chan, Minesh P. Mehta, Ying Xiao

**Affiliations:** 1Department of Radiation Oncology, University of Pennsylvania, Philadelphia, PA 19104, USAying.xiao@pennmedicine.upenn.edu (Y.X.); 2Northwestern Medicine Cancer Center Warrenville, Warrenville, IL 60555, USA; 3Memorial Sloan Kettering Cancer Center, New York, NY 10065, USA; 4McGill University Health Centre, Montreal, QC H4A 3J1, Canada; 5Cleveland Clinic Foundation, Cleveland, OH 44195, USA; 6Department of Radiology, University of Michigan, Ann Arbor, MI 48109, USA; tlchenev@umich.edu (T.L.C.);; 7Department of Radiation Oncology, Washington University School of Medicine, St. Louis, MO 63110, USA; 8National Cancer Institute, Bethesda, MD 20892, USA; 9Stanford Cancer Institute, Stanford, CA 94305, USA; qle@stanford.edu (Q.-T.L.);; 10Saint Joseph’s Hospital and Medical Center, Phoenix, AZ 85013, USA; 11UPMC-Shadyside Hospital, Case Western Reserve University, Pittsburgh, PA 15232, USA; 12Department of Radiation Oncology, Thomas Jefferson University Hospital, Philadelphia, PA 19107, USA; 13Stephenson Cancer Center, University of Oklahoma Health Sciences Center, Oklahoma City, OK 73104, USA; 14UT Southwestern, Simmons Cancer Center, Dallas, TX 75235, USA; 15Baptist Comprehensive Cancer Center, Wake Forest University Health Sciences, Winston-Salem, NC 27157, USA; 16Miami Cancer Institute, Miami, FL 33176, USA

**Keywords:** radiotherapy, knowledge-based planning, brain and head/neck cancers, clinical trial quality assurance

## Abstract

**Simple Summary:**

Radiation therapy (RT) plans deviating from the standard could likely compromise the efficacy of a pre-specified intervention for any clinical trial due either to insufficient coverage of the target area and/or excessive radiation doses to healthy tissues. Knowledge-based machine learning tools utilize high-quality data and generate patient-specific optimization objectives that produce RT plans that comply better with treatment protocol specifications. In this study, we investigated the use of a knowledge-based planning (KBP) model to evaluate the quality of RT plans in two clinical trials, one for glioblastoma and the other for head and neck cancer. The outcomes of this research indicate that this tool can assist RT quality assessments in multi-center clinical trials.

**Abstract:**

The quality of radiation therapy (RT) treatment plans directly affects the outcomes of clinical trials. KBP solutions have been utilized in RT plan quality assurance (QA). In this study, we evaluated the quality of RT plans for brain and head/neck cancers enrolled in multi-institutional clinical trials utilizing a KBP approach. The evaluation was conducted on 203 glioblastoma (GBM) patients enrolled in NRG-BN001 and 70 nasopharyngeal carcinoma (NPC) patients enrolled in NRG-HN001. For each trial, fifty high-quality photon plans were utilized to build a KBP photon model. A KBP proton model was generated using intensity-modulated proton therapy (IMPT) plans generated on 50 patients originally treated with photon RT. These models were then applied to generate KBP plans for the remaining patients, which were compared against the submitted plans for quality evaluation, including in terms of protocol compliance, target coverage, and organ-at-risk (OAR) doses. RT plans generated by the KBP models were demonstrated to have superior quality compared to the submitted plans. KBP IMPT plans can decrease the variation of proton plan quality and could possibly be used as a tool for developing improved plans in the future. Additionally, the KBP tool proved to be an effective instrument for RT plan QA in multi-center clinical trials.

## 1. Introduction

Glioblastoma is the most aggressive primary malignant brain tumor found in humans, with 5-year overall survival less than 7% after surgery and conventional chemoradiotherapy [1,2]. The conventional therapeutic regimen for GBM comprises resection to the greatest extent safely feasible, followed by concurrent and adjuvant temozolomide (TMZ) chemotherapy and RT [3,4,5]. Despite enhancements in patient survival with this combinatorial approach, local disease control remains a challenge and a major cause of therapeutic failure [6]. In an endeavor to improve outcomes, NRG Oncology initiated the phase II randomized trial NRG-BN001 to assess the impact of escalated RT doses in conjunction with TMZ for GBM patients. This trial also aims to compare the benefits of proton beam therapy versus photon IMRT, potentially enabling higher RT doses without escalating toxicity and specifically diminishing the risk of lymphopenia, which is supported by level 1 evidence. Indirect data suggest reduced survival with more severe lymphopenia.

Nasopharyngeal carcinoma (NPC) poses significant challenges for RT planning due to the complexity of planning target volumes (PTVs), the necessity for simultaneous integrated boost techniques, and the imperative of sparing multiple OARs [7,8]. NRG-HN001, a phase II/III multi-institutional clinical trial, targets patients diagnosed with NPC to investigate and optimize therapeutic strategies. In addition, both NRG-BN001 and NRG-HN001 represent pivotal investigations into the application of proton therapy within phase III clinical trials. The results of these trials are anticipated to provide evidence regarding whether proton therapy can positively influence patient outcomes, specifically in terms of augmenting efficacy and/or decreasing toxicities.

The correlation between adherence to established guidelines in RT treatment planning and clinical outcomes is well-documented; deviations from such protocols are linked with diminished survival rates, increased probability of disease progression, and a greater risk of RT-induced complications. Consequently, rigorous QA of RT plans is a pivotal QA component for clinical trials that incorporate RT [9,10,11].

The Imaging Radiation Oncology Core (IROC) of the National Clinical Trials Network has conducted QA reviews of all treatment plans of patients enrolled in NRG Oncology clinical trials. Although this process can easily identify plans that deviate from protocol-defined criteria, it does not capture the intricacies and challenges inherent in individual patient plans. Moreover, the current IROC QA process does not offer insights or possibilities for enhancing the quality of the treatment plans. 

The rapid development of Artificial Intelligence (AI) in recent years offers a promising solution to these challenges. Knowledge-based planning (KBP) is a specialized application of AI designed to improve radiation therapy planning by using historical data to build predictive models. These models, trained on high-quality, protocol-compliant plans, learn the optimal dosimetric parameters based on patient geometry, enabling the creation of customized radiation therapy (RT) plans [12,13]. KBP significantly reduces the variability that often arises in RT planning from differences in planner experience and institutional practices [14,15,16,17]. By providing a data-driven benchmark for plan quality, KBP promotes consistency and efficiency in plan evaluation across multiple treatment centers, which is particularly valuable in multi-center clinical trials.

Although several recent publications have shown the feasibility of KBP-assisted intensity-modulated radiation therapy (IMRT) treatment planning in clinical settings [17,18,19,20,21,22], implementing knowledge-based proton planning in clinical trial evaluation is in its infancy [23]. Prior research has illuminated the application of the KBP model in evaluating the quality of photon plans submitted to NRG-HN001. Building on this foundation, our current study aims to extend the evaluation framework to include both photon and proton treatment plans submitted to the NRG-BN001 trial as well as to assess the quality of NRG-HN001 IMPT plans utilizing a knowledge-based approach. This comprehensive assessment seeks to leverage the insights gleaned from KBP models to ensure and enhance the quality of treatment plans across different modalities and clinical scenarios.

## 2. Materials and Methods

### 2.1. Patient Cohort

This research incorporated a study population of 203 patients diagnosed with glioblastoma (GBM) who were part of the NRG-BN001 clinical trial and 70 patients with nasopharyngeal carcinoma (NPC) who participated in the NRG-HN001 study. In the context of the NRG-BN001 trial, 139 patients received photon therapy, and 64 patients received proton therapy, with both groups undergoing dose-intensified radiotherapy with a simultaneous Integrated Boost. This latter group was further divided into 36 cases treated with IMPT and 28 cases undergoing passive scattered (PS) proton therapy. All participants in this trial were prescribed 50 Gy (relative biological effectiveness [RBE] for protons) in 30 fractions to the FLAIR or T2 abnormality, with a simultaneous integrated boost to 75 Gy ([RBE] for protons) to the postoperative cavity and residual enhancing disease. Regarding the NRG-HN001 trial, there were 50 patients treated with IMRT and 20 patients treated with proton therapy. The prescribed dose was either 69.96 Gy in 33 fractions or 70 Gy in 35 fractions. Both trials set forth specific dosimetric compliance standards. These standards, relevant to both targets and OARs, are elaborated in Appendix A, respectively. Any structures not complying with the protocol’s accepted variation thresholds are categorized as unacceptable deviations.

### 2.2. Knowledge-Based Planning and Model Configuration

#### 2.2.1. Photon Model

Fifty per-protocol IMRT plans from the NRG-BN001 photon group were chosen to develop the photon RapidPlan^®^ RT (Varian Medical System, Palo Alto, CA, USA) model. The initial parameters of the model were established in accordance with the priorities provided in the protocol. Structures assigned a higher priority level were given a greater priority value. These same plans also served as an internal validation cohort, aiding in the refinement of model parameters. The final KBP photon model’s defined objective list is detailed in Appendix A. For the re-optimization of photon plans, the Photon Optimizer (PO) for IMRT (version 16.0.2), the Dose-Volume Histogram (DVH) Estimation Algorithm (version 16.0.2) for DVH estimation, and the Anisotropic Analytical Algorithm (AAA, version 16.0.2) for volume and portal dose computation were selected as the calculation models.

#### 2.2.2. Proton Models

For each trial, fifty patients enrolled in photon cohorts were manually re-planned with the IMPT technique using golden beam data of the ProBeam proton therapy system. The volume dose was calculated based on the Proton Convolution Superposition algorithm (PCS, v. 16.0.2) with a 5 mm spot size and 2.5 mm resolution. The fluence-based Nonlinear Universal Proton Optimizer (NUPO, v. 16.0.2) and the multifield simultaneous spot optimization method were applied to optimize dose distribution. 

The 50 manually generated IMPT plans were evaluated based on the dosimetric compliance criteria specified in each protocol and were subsequently utilized to train the preliminary models. To enhance the performance of these models, a closed-loop iteration was implemented by re-optimizing the library IMPT plans using the initial RP model and updating the model with the re-optimized cases. For NRG-BN001, the KBP IMPT models also include three control regions aimed at optimizing the dose distribution. These additional regions comprised the Planning Risk Volume (PRV), defined as PTV_5000 minus (PTV_7500 plus a 5 mm margin); PTV_5000 opt, delineated as PTV_5000 excluding PTV_7500; and a ‘ring’ region, specified as a 1 cm margin encircling PTV_5000. Appendix A list the optimization objectives and priorities specified in the final KBP proton model for NRG-BN001 and NRG-HN001, respectively.

### 2.3. Plan Evaluation

The plan quality of the submitted photon and proton plans was assessed through comparison with the KBP photon and proton plans. The evaluation was conducted based on protocol compliance, target dose conformality index (CI) and homogeneity index (HI), and the dosimetric endpoints, including PTVs and critical structures [24,25]. IROC has developed a QA workflow to evaluate RT plans. This systematic approach assesses adherence to the protocol-defined dose constraints and categorizes RT plans into three distinct scores: per-protocol: score 1, variation acceptable: score 2, and deviation unacceptable: score 3.

The conformality index was calculated based on the Paddick index [26], defined as: (1)$$CI=\frac{{TV}_{PIV}^{2}}{PIV\times TV}$$
where ${TV}_{PIV}$ is the target volume encompassed by the prescription isodose, PIV is the prescription isodose volume, and TV is the target volume. The homogeneity index was defined as the ratio of the maximum point dose $D_{max}$ and the prescribed dose $D_{Rx}$ [27]:(2)$$HI=\frac{D_{max}}{D_{Rx}}$$

To evaluate the differences in quality between the plans submitted initially and those derived from KBP, mean dosimetric parameters were assessed. Furthermore, a paired T-test was employed to conduct a statistical comparison. 

## 3. Results

### 3.1. NRG-BN001 Photon Plan Quality Review

Table 1 lists the results of the 139 photon plans submitted to NRG-BN001 using the IROC QA procedure before and after KBP model optimization. The KBP plans show substantially better quality; the number of cases that failed to meet the per-protocol and variation acceptable criteria dropped by 39% and 60.1%, respectively.

Table 2 presents a detailed dosimetric comparison between the initially submitted intensity-modulated radiation therapy (IMRT) plans and the KBP plans. On average, both groups of plans demonstrate satisfactory target coverage, adhering to the specified protocol constraints for all anatomical structures. There is a notable equivalence in target dose coverage (PTV_7500 D95%[Gy]: ∆ = 0.2 ± 1.9 Gy), conformality index (∆ = 0.0 ± 0.20), and the homogeneity index (∆ = −0.02 ± 0.03) between the submitted and KBP plans. The application of KBP is particularly advantageous for organs of higher priority, as it facilitates a reduction in dosage. This improvement is evident in the spinal cord (∆ = −0.9 ± 3.0), brain stem_core/surf (∆ = −1.7 ± 6.2 and −2.2 ± 5.9), optic chiasm_PRV (∆ = −3.6 ± 6.8), and optic nerve_PRV (∆ = −2.6 ± 5.8). Figure 1 further illustrates this outcome, showcasing a dose wash comparison for an exemplary case from the photon cohort between the submitted and KBP plans. 

### 3.2. NRG-BN001 Proton Plan Quality Review

Table 3 presents the IROC QA review results for the 64 proton plans submitted to NRG-BN001 and the KBP IMPT plans. The KBP plans show substantially better quality; the number of cases that failed to meet the per-protocol and variation acceptable criteria was reduced by 77.6% and 66.7%, respectively. 

Table 4 methodically outlines the average variations in pivotal dosimetric parameters between the originally submitted proton plans and the KBP IMPT plans. Notably, the maximum dose imparted to the brain stem_core, brain stem_surf, optic chiasm_PRV, optic nerve_PRV, and retina demonstrated a significant reduction (*p* < 0.05) in the KBP IMPT plans. The reductions were quantified as 10.3 ± 6.1 Gy, 12.8 ± 8.2 Gy, 10.6 ± 10.2 Gy, 4.3 ± 5.3 Gy, and 4.2 ± 6.9 Gy for the IMPT group and 9.7 ± 9.6 Gy, 10.3 ± 8.6 Gy, 12.8 ± 13.2 Gy, 5.4 ± 8.0 Gy, and 4.1 ± 8.2 Gy for the PS group. Figure 2 provides a visual representation of the dose distribution for a typical case within the proton cohort. In comparison to the clinically submitted plan, the KBP IMPT plan exhibits significant enhancement in both target coverage and OAR sparing. However, it is noteworthy that the KBP plan is associated with an elevated maximum dose to the PTV and an increased volume of brain tissue subjected to radiation exposure.

### 3.3. NRG-HN001 Proton Plan Quality Review

In Table 5, the results from the IROC QA review of 20 proton therapy plans submitted to the NRG-HN001 are compared with KBP IMPT plans. The KBP IMPT plans demonstrated a notable improvement in compliance with the protocol criteria, with a reduction in the number of non-compliant cases by 54.8% and 75%, respectively.

Table 6 presents an analysis of the average differences in critical dosimetric parameters between the submitted proton plans and the KBP IMPT plans. The analysis revealed that the KBP IMPT plans achieved a significant reduction in the maximum doses delivered to various critical structures. Specifically, reductions were observed in the brainstem (4.6 ± 6.8 Gy, *p* = 0.008), optic chiasm (8.9 ± 12 Gy, *p* = 0.040), left optic nerve (9.8 ± 10 Gy, *p* = 0.034), right optic nerve (13.1 ± 10 Gy, *p* = 0.002), left temporomandibular joint (4.6 ± 7.2 Gy, *p* = 0.195), right temporomandibular joint (5.6 ± 6.6 Gy, *p* = 0.146), left parotid gland (2.9 ± 5 Gy, *p* = 0.040), and right parotid gland (2.5 ± 4.7 Gy, *p* = 0.067). 

Figure 3 provides a visual comparison of dose distribution in a typical case from the proton plan cohort. The KBP IMPT plans demonstrate superior target coverage and OAR sparing, although it is noteworthy that the KBP plan resulted in a higher maximum dose to the PTV and an increased volume receiving the prescription dose compared to the submitted clinical plan.

## 4. Discussion

This investigation revealed more pronounced improvements in adherence to treatment protocols using the KBP model across both photon and proton modalities. The proton cohort showed notably superior dosimetric enhancements, with dose reductions ranging from 1.1 to 12.8 Gy, compared to the photon cohort, which observed improvements between 1.1 and 3.6 Gy. Although the examined photon plans displayed high quality—reflecting a mature development and implementation of IMRT techniques—the findings suggest there are still considerable opportunities for improvement through KBP. This could potentially refine treatment outcomes and increase adherence to established dosimetric guidelines.

The superior performance of KBP proton plans can be attributed to the use of specific advanced technologies, particularly the Varian Eclipse treatment planning system and the Varian ProBeam beam model. These results demonstrate that treatment outcomes can vary significantly depending on the technology and software used, as well as the level of expertise in treatment planning across different institutions. This study not only confirms the effectiveness of KBP in optimizing radiation therapy plans but also highlights the crucial role of technological and professional expertise in achieving optimal therapeutic outcomes. It emphasizes the need for continuous advancements in treatment planning tools and methodologies along with ongoing professional development and training for clinicians to fully harness the therapeutic potential of proton therapy.

Additionally, this study highlights the value of KBP tools as a robust benchmark for quality assurance in radiation therapy plans submitted for clinical trials. The future integration of KBP into the quality assurance processes of clinical trials could be pivotal, enhancing workflow efficiency and enabling a more thorough evaluation of treatment plans.

In this study, the evaluated proton plans submitted to NRG-BN001 incorporated both IMPT and PS plans. Between 2015 and 2016, a predominant majority (81%) of cases were treated with PS proton therapy, whereas this ratio decreased to 16% for patients enrolled between 2017 and 2019. In addition, the submitted IMPT plans had considerably superior dose conformity to the target than the submitted PS plans (CI PTV_7500: 0.80 ± 0.16 vs. 0.68 ± 0.31). Despite the motion insensitivity of the PS system and less complex beam delivery [28], components such as scatter foils, range modulator wheel, and patient-specific compensator restrict beam conformity and limit maximum treatment depth [29]. In contrast, the pencil beam scanning (PBS) system uses magnets to manipulate the proton beam, enabling superior conformity, deeper treatment, and less neutron generation, as scattering foils and compensators are not required. The utilization of the PBS system, however, presents challenges such as increased complexity, longer delivery time, and low motion tolerance [30,31,32]. Consequently, while the passive scattering system was initially predominant, the pencil beam scanning approach is becoming progressively favored [32].

Recent studies have demonstrated the viability of using a KBP model for treatment plan QA and re-optimization [33,34,35]. The utilization of KBP has been shown to elevate planning efficiency and quality with reduced variability, thereby serving as a robust benchmark for clinical plans. The integration of these models within clinical trial QA processes not only enhances the overall quality but also has the potential to abbreviate the learning curve for clinical practitioners [14,25,36,37]. In our analysis, the application of these models has been particularly instrumental in evaluating the quality of treatment plans submitted to clinical trials, underscoring their utility as a critical assessment tool.

This study acknowledges several limitations. The constrained availability of proton plans necessitated the reliance on IMPT plans reconstituted from photon cases as surrogates for training the KBP proton models. This approach could introduce a potential bias into the training library, possibly affecting the model’s quality. Moreover, the KBP models were constructed using IMPT plans derived from golden beam models provided by the system manufacturer, which might not align with the capabilities of equipment at participating institutions. Therefore, the plan quality depicted in this analysis represents what is attainable under ideal conditions with specific beam models and techniques, rather than a definitive representation of the variability in quality that might be encountered in real-world practice. Additionally, our premise is that improved plan quality could positively influence patient outcomes by reducing side effects and improving overall quality of life. However, it is important to note that as the clinical trials assessed in this study are still ongoing, we do not currently have access to the outcome data necessary to directly evaluate these potential benefits.

## 5. Conclusions

This investigation evaluated the quality of RT plans submitted to the multi-institutional clinical trials NRG-BN001 and NRG-HN001. It demonstrates the efficacy of KBP models in generating protocol-compliant plans and for RT plan QA. The findings indicate that the photon plans submitted to the NRG-BN001 clinical trial generally exhibit commendable quality. However, there is a marked variability in the quality of proton plans submitted for both NRG-BN001 and NRG-HN001, highlighting the emerging nature of this therapy. This study introduced the KBP-based models, which serve as a benchmark for the quality of plans that can be achieved in the management of tumors of the brain and head/neck region with radiation therapy. The KBP models built in this study will be published and made accessible to both the research and clinical communities for the purpose of RT plan QA and optimization.

## Figures and Tables

**Figure 1 cancers-16-02007-f001:**
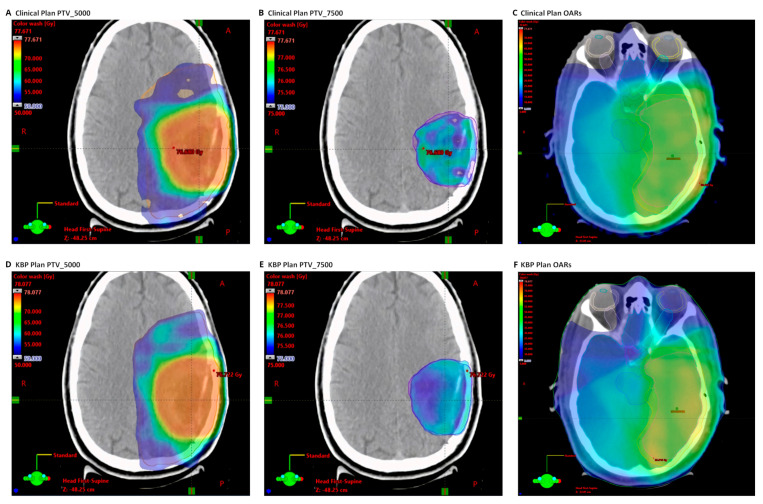
Dose distribution of an example case in the photon cohort. The clinical plan was the originally submitted plan and the KBP plan was generated using the IMRT RapidPlan model. (**A**–**C**) Dose distribution of the clinical plan. (**D**–**F**) Dose distribution of the KBP plan. The KBP plan demonstrates enhanced sparing of OARs with higher priority, including the brainstem, optic chiasm, and optic nerve.

**Figure 2 cancers-16-02007-f002:**
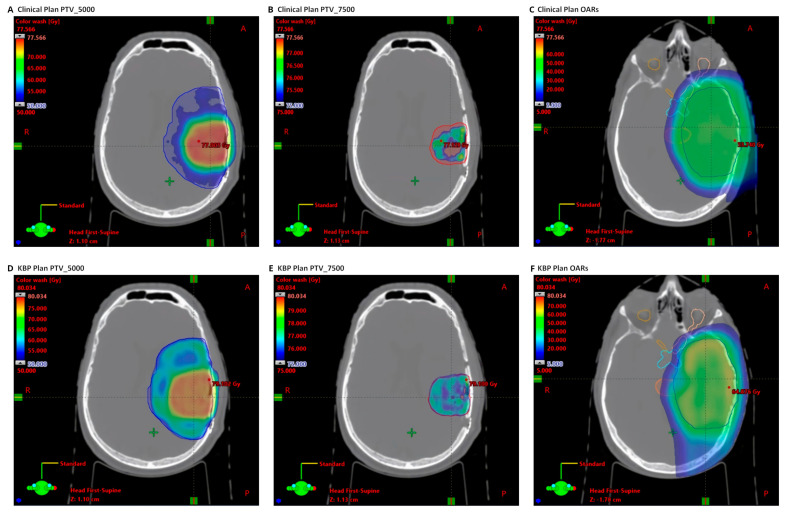
Dose distribution of an example proton therapy case. Panels (**A**–**C**) illustrate the dose distribution of the clinical plan, while panels (**D**–**F**) show the dose distribution of the KBP plan. The KBP plan demonstrates enhanced target coverage and reduced dose delivered to adjacent OARs, including the brainstem, optic chiasm_PRV, and left optic nerve_PRV.

**Figure 3 cancers-16-02007-f003:**
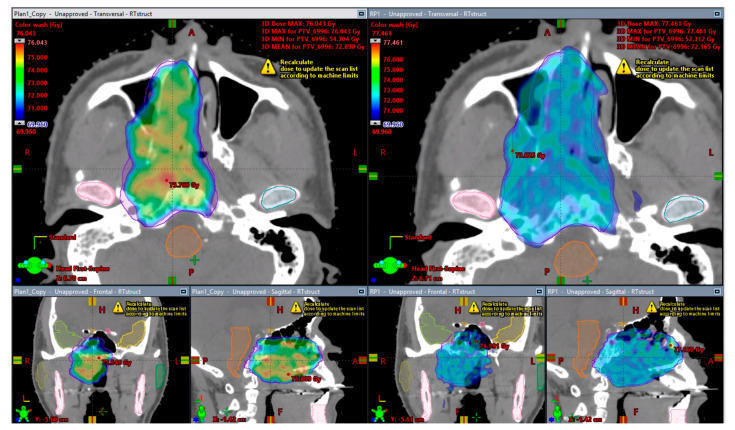
Dose distribution of an example case in the proton cohort of NRG-HN001. KBP IMPT plan (**right**) versus original submitted IMPT plan (**left**). KBP plan demonstrates enhanced target coverage and better sparing of the right temporal lobe.

**Table 1 cancers-16-02007-t001:** Comparison of IROC QA scores between the submitted NRG-BN001 photon plans and the KBP plans.

Structures	Dosimetric Parameter	Photon Submitted			Photon KBP		
		Score 1	Score 2	Score 3	Score 1	Score 2	Score 3
PTV_5000	D95%[Gy]	110	27	2	130	9	0
PTV_7500	D95%[Gy]	105	26	8	111	19	9
PTV_7500	D10%[Gy]	101	35	3	138	1	0
PTV_7500	D0.03cc[Gy]	84	47	8	126	13	0
SpinalCord	D0.03cc[Gy]	139	0	0	139	0	0
BrainStemCore	D0.03cc[Gy]	125	13	1	126	13	0
BrainStemSurf	D0.03cc[Gy]	121	18	0	124	15	0
OpticChiasm_PRV	D0.03cc[Gy]	129	10	0	130	9	0
OpticNerve_L_PRV	D0.03cc[Gy]	135	3	1	134	5	0
OpticNerve_R_PRV	D0.03cc[Gy]	133	5	1	132	6	1
Retina_L	D0.03cc[Gy]	138	1	0	139	0	0
Retina_R	D0.03cc[Gy]	139	0	0	137	2	0
Brain	D5% [Gy]	136	2	1	139	0	0
Lens_L	D0.03cc[Gy]	126	11	2	118	20	1
Lens_R	D0.03cc[Gy]	126	12	1	123	16	0

Score 1: per-protocol; Score 2: variation acceptable; Score 3: deviation unacceptable.

**Table 2 cancers-16-02007-t002:** Dosimetric comparison of submitted and KBP photon plans for NRG-BN001.

Structures	Dosimetric Parameter	Photon Submitted	Photon KBP	*p* Value
PTV_5000	D95%[Gy]	50.9 ± 1.5	51.7 ± 1.2	<0.001 *
PTV_7500	D95%[Gy]	74.2 ± 2.9	74.0 ± 2.8	0.001 *
PTV_7500	D10%[Gy]	78.1 ± 1.6	76.7 ± 0.8	<0.001 *
PTV_7500	D0.03cc[Gy]	79.8 ± 1.9	78.5 ± 1.3	<0.001 *
Spinal Cord	D0.03cc[Gy]	6.6 ± 5.9	5.6 ± 5.1	<0.001 *
BrainStem_Core	D0.03cc[Gy]	46.6 ± 11.6	44.6 ± 14.0	0.019 *
BrainStem_Surf	D0.03cc[Gy]	46.1 ± 12.7	43.6 ± 14.7	<0.001 *
OpticChiasm_PRV	D0.03cc[Gy]	39.7 ± 16.3	35.9 ± 16.9	<0.001 *
OpticNerve_PRV	D0.03cc[Gy]	33.7 ± 20.1	31.1 ± 20.2	<0.001 *
Retina	D0.03cc[Gy]	16.4 ± 12.3	17.5 ± 12.1	0.054
Brain	D5% [Gy]	72.1 ± 7.0	72.42 ± 6.2	0.803
Lens	D0.03cc[Gy]	4.6 ± 2.7	5.3 ± 2.6	<0.001 *
HI_PTV_7500_		1.06 ± 0.02	1.04 ± 0.02	<0.001 *
CI_PTV_7500_		1.00 ± 0.18	1.00 ± 0.10	0.39

The paired *t*-test was employed; asterisks (*) indicate a statistically significant difference between the submitted and KBP plan.

**Table 3 cancers-16-02007-t003:** Comparison of IROC QA scores between the submitted NRG-BN001 proton plans and the KBP plans.

Structures	Dosimetric Parameter	Proton Submitted			Proton KBP		
		Score 1	Score 2	Score 3	Score 1	Score 2	Score 3
PTV_5000	D95%[Gy]	49	14	1	63	1	0
PTV_7500	D95%[Gy]	37	22	5	61	1	2
PTV_7500	D10%[Gy]	60	4	0	64	0	0
PTV_7500	D0.03cc[Gy]	63	1	0	57	7	0
SpinalCord	D0.03cc[Gy]	64	0	0	64	0	0
BrainStemCore	D0.03cc[Gy]	61	3	0	64	0	0
BrainStemSurf	D0.03cc[Gy]	58	6	0	63	1	0
OpticChiasm_PRV	D0.03cc[Gy]	62	2	0	63	1	0
OpticNerve_L_PRV	D0.03cc[Gy]	61	3	0	64	0	0
OpticNerve_R_PRV	D0.03cc[Gy]	62	1	1	62	1	1
Retina_L	D0.03cc[Gy]	64	0	0	64	0	0
Retina_R	D0.03cc[Gy]	63	0	1	64	0	0
Brain	D5%[Gy]	64	0	0	64	0	0
Lens_L	D0.03cc[Gy]	63	0	1	64	0	0
Lens_R	D0.03cc[Gy]	62	2	0	63	1	0

Score 1: per-protocol; Score 2: variation acceptable; Score 3: deviation unacceptable.

**Table 4 cancers-16-02007-t004:** Comparison of dosimetric parameters in submitted and KBP proton plans of BN001 clinical trial.

Structures	Dosimetric Parameter	IMPT			PS		
		Clinical	KBP	*p* Value	Clinical	KBP	*p* Value
PTV_5000	D95%[Gy]	50.9 ± 1.7	50.8 ± 0.4	0.772	50.3 ± 1.5	50.9 ± 0.35	0.037 *
PTV_7500	D95%[Gy]	73.3 ± 3.5	75.0 ± 1.4	<0.001 *	74.2 ± 1.5	75.2 ± 0.5	0.002 *
PTV_7500	D10%[Gy]	77.4 ± 0.8	77.5 ± 0.4	0.22	77.1 ± 1.5	77.5 ± 0.4	0.164
PTV_7500	D0.03cc[Gy]	78.4 ± 1.0	79.5 ± 0.8	<0.001 *	78.4 ± 1.8	79.3 ± 0.6	0.015*
Spinal Cord	D0.03cc[Gy]	0.1 ± 0.1	0.0 ± 0.0	0.005 *	0.1 ± 0.3	0.0 ± 0.0	0.364
BrainStem_Core	D0.03cc[Gy]	44.0 ± 14.7	33.6 ± 15.1	<0.001 *	38.3 ± 17.0	28.6 ± 14.2	<0.001 *
BrainStem_Surf	D0.03cc[Gy]	43.0 ± 16.3	30.2 ± 16.4	<0.001 *	38.8 ± 19.3	28.5 ± 14.3	<0.001 *
OpticChiasm_PRV	D0.03cc[Gy]	31.7 ± 22.2	21.1 ± 20.6	<0.001 *	34.8± 20.6	22.0 ± 18.4	<0.001 *
OpticNerve_PRV	D0.03cc[Gy]	24.3 ± 25.6	20.0 ± 22.7	<0.001 *	22.0 ± 23.2	16.5 ± 19.7	0.002 *
Retina	D0.03cc[Gy]	7.4 ± 13.6	3.2 ± 8	<0.001 *	7.5 ± 12.8	3.3 ± 7.0	0.014 *
Brain	D5%[Gy]	71.4 ± 5.0	72.6 ± 4.6	0.013 *	73.2 ± 5.6	72.3 ± 5.3	0.059
Lens	D0.03cc[Gy]	1.2 ± 2.8	0.4 ± 1.5	0.027 *	1.7 ± 0.4	0.7 ± 0.3	0.689
HI_PTV_7500_		1.05 ± 0.01	1.06 ± 0.01	<0.001 *	1.06 ± 0.01	1.04 ± 0.02	<0.008 *
CI_PTV_7500_		0.80 ± 0.16	0.89 ± 0.07	0.002 *	0.68 ± 0.31	0.89 ± 0.08	0.003 *

The paired *t*-test was employed; asterisks (*) indicate a statistically significant difference between the submitted and KBP plan.

**Table 5 cancers-16-02007-t005:** Comparison of IROC QA scores between the submitted NRG-HN001 proton plans and the KBP plans.

Structures	Dosimetric Parameter	Proton Submitted			Proton KBP		
		Score 1	Score 2	Score 3	Score 1	Score 2	Score 3
PTV_High	V100%[%]	3	4	13	13	4	3
PTV_High	D99%[%]	6	3	11	12	2	6
PTV_High	D0.03cc[%]	20	0	0	18	2	0
PTV_Intermediate1	V63Gy[%]/V62.7Gy[%]	2	0	7	8	1	0
PTV_Intermediate2	V59Gy[%]/V59.4Gy[%]	1	3	5	8	1	0
PTV_Low	V56Gy[%]	1	2	8	9	0	2
SpinalCord	D0.03cc[Gy]	18	2	0	20	0	0
BrainStem	D0.03cc[Gy]	13	7	0	20	0	0
OpticChiasm	D0.03cc[Gy]	17	1	0	20	0	0
OpticNerve_L	D0.03cc[Gy]	17	1	0	20	0	0
OpticNerve_R	D0.03cc[Gy]	17	1	0	20	0	0
TMjoint_L	D0.03cc[Gy]	15	0	0	15	0	0
TMjoint_R	D0.03cc[Gy]	15	0	0	15	0	0
Mandible	D0.03cc[Gy]	19	1	0	19	1	0
BrachialPlexus_L	D0.03cc[Gy]	18	2	0	20	0	0
BrachialPlexus_R	D0.03cc[Gy]	20	0	0	19	1	0
TemporalLobe_L	D0.03cc[Gy]	19	1	0	19	1	0
TemporalLobe_R	D0.03cc[Gy]	20	0	0	20	0	0
Parotid_L	Mean[Gy]	17	2	1	20	0	0
Parotid_R	Mean[Gy]	16	1	3	18	1	1

Score 1: per-protocol; Score 2: variation acceptable; Score 3: deviation unacceptable.

**Table 6 cancers-16-02007-t006:** Comparison of dosimetric parameters in submitted and KBP proton plans from HN001 clinical trial.

Structures	Dosimetric Parameter	Proton Submitted	Proton KBP	*p* Value
PTV_High	V100%[%]	42.1% ± 41.8%	84.1% ± 30.4%	<0.001 *
PTV_High	D99%[%]	89.7% ± 5.4%	93.3% ± 6.7%	0.037 *
PTV_High	D0.03cc[%]	102.2% ± 4.0%	111.0% ± 4.9%	<0.001 *
PTV_Intermediate1	V63Gy[%]/V62.7Gy[%]	53.3% ± 29.7%	97.1% ± 2.0%	<0.001 *
PTV_Intermediate2	V59Gy[%]/V59.4Gy[%]	77.3% ± 21.2%	97.1% ± 2.5%	<0.001 *
PTV_Low	V56Gy[%]	63.8% ± 24.7%	91.4% ± 15.0%	0.003 *
SpinalCord	D0.03cc[Gy]	37 ± 10.5	37.1 ± 2.3	0.486
BrainStem	D0.03cc[Gy]	51.3 ± 7.4	46.7 ± 2.9	0.008 *
OpticChiasm	D0.03cc[Gy]	34.3 ± 14.7	25.4 ± 15.7	0.040 *
OpticNerve_L	D0.03cc[Gy]	41.7 ± 14.3	31.9 ± 17.6	0.034 *
OpticNerve_R	D0.03cc[Gy]	42.7 ± 11.3	29.6 ± 15.2	0.002 *
TMjoint_L	D0.03cc[Gy]	52.5 ± 11.1	47.8 ± 16.4	0.195
TMjoint_R	D0.03cc[Gy]	50 ± 11.4	44.3 ± 16	0.146
Mandible	D0.03cc[Gy]	63.6 ± 6.7	64.3 ± 4.7	0.361
BrachialPlexus_L	D0.03cc[Gy]	60.9 ± 4.3	61.3 ± 2.3	0.345
BrachialPlexus_R	D0.03cc[Gy]	60.7 ± 3.6	61.6 ± 2.8	0.208
TemporalLobe_L	D0.03cc[Gy]	61.6 ± 6.8	60.7 ± 7.9	0.438
TemporalLobe_R	D0.03cc[Gy]	62.6 ± 6.2	62.3 ± 7.4	0.342
Parotid_L	Mean[Gy]	25.5 ± 6.3	22.6 ± 5.4	0.040 *
Parotid_R	Mean[Gy]	24.5 ± 5.3	22.1 ± 2.8	0.067
HIPTV_high		1 ± 0	1.1 ± 0	<0.001 *
CI PTV_High		0.3 ± 0.3	0.7 ± 0.3	<0.001 *

The paired *t*-test was employed; asterisks (*) indicate a statistically significant difference between the submitted and KBP plan.

## Data Availability

Third-party data: Restrictions apply to the availability of these data. Data were obtained from Imaging Oncology Core Radiotherapy Quality Assurance team. This is an ongoing trial; no data will be made available to the public before any publication of the endpoint of this trial. After the closure and publication of the endpoint of this trial, data can be requested via data sharing through NRG Oncology.

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
