# Peer review of "Radiotherapy Plan Quality Assurance in NRG Oncology Trials for Brain and Head/Neck Cancers: An AI-Enhanced Knowledge-Based Approach"

_cancers, 2024, doi:10.3390/cancers16112007_

Round 1
Reviewer 1 Report
Comments and Suggestions for Authors
This study assessed radiation therapy (RT) plan quality using knowledge-based planning (KBP) models in multi-center clinical trials for glioblastoma (NRG-BN001) and head/neck cancer (NRG-HN001). KBP tools leverage machine learning to optimize RT plans, ensuring protocol compliance and target coverage while minimizing healthy tissue exposure. Evaluation of 203 glioblastoma and 70 nasopharyngeal carcinoma patients revealed superior quality in KBP-generated plans compared to submitted plans. Notably, proton plans to exhibit variable quality, underscoring the evolving nature of proton therapy. The study introduces a benchmark KBP-based intensity-modulated proton therapy (IMPT) model, enhancing RT plan quality assurance and optimization for brain and head/neck tumors. Published KBP models will benefit research and clinical communities by facilitating robust RT plan QA. The topic and results of the manuscript are important. The manuscript can be published after minor modifications.
1. The title of the article is too wordy. Consider revising it.
2. Figures 1 and 2 refer to components labeled (A-C) and (D-F) in their captions, but these labels are not present directly on the images themselves. Update the images to visibly display the indicated labels (A-C) and (D-F).
3. Add key findings from the results into the figure captions (Figures 1 – 3) to enhance understanding of the differences achieved between the Clinical Plan and KBP Plan.
4. The discussion section requires improvements outlining the importance of observed results and future applications.
5. Explain the importance of using KBP-based IMPT models as a benchmark for RT plan quality in enhancing the effectiveness of treatment. Explain in the manuscript.
6. I recommend thoroughly reviewing the manuscript to identify and rectify any typos and grammatical errors.
Comments on the Quality of English LanguageI recommend thoroughly reviewing the manuscript to identify and rectify any typos and grammatical errors.
Reviewer 2 Report
Comments and Suggestions for Authors
1. Line 228, 246, 267 - whenever mentioned the word “significant” – should also reporting the corresponding p-value to support such statement.
2. What were the citations/reference for authors to choose these evaluation criteria? Line 154 – “The evaluation was conducted based on protocol compliance, target dose conformality index (CI) and homogeneity index (HI), and the dosimetric endpoints including PTVs and critical structures. “
3. Besides descriptive data listed in Table 1/Table 3/Table 5, authors should conduct statistical analysis to prove Photon KBP was better.
4. Table 2/Table 4 /Table 6 – should add footnote for p-value mentioned what kind of statistical test performed.
5. Lack multivariable statistical model to prove the Photon KBP was better.
6. Suggested to add a section mention the limitation of the method/study.
7. What were the major patients benefits of this knowledge-based planning (KBP) in RT? Reduced side effects? Improve quality of life? If not, then these are the limitations of this method/study.
Reviewer 3 Report
Comments and Suggestions for Authors
This study evaluated the quality of radiation therapy (RT) treatment plans for brain and head/neck cancers within multi-institutional clinical trials using Knowledge-Based Planning (KBP) techniques. The research focused on 203 Glioblastoma (GBM) patients in NRG-BN001 and 70 nasopharyngeal carcinoma (NPC) patients in NRG-HN001. The study suggests that KBP is an effective tool for quality assurance (QA) in RT plans across diverse clinical settings. The study addresses an important subject and presents significant findings. I recommend accepting the manuscript with minor revisions.
1. Revise the title of the manuscript.
2. The background information on the subject needs to be explained more in the introduction section explaining how AI can be helpful to the studies in this domain.
3. Figures labels are missing in this manuscript.
4. The manuscript contained several grammatical errors and typos. Ensure thorough revision to correct these issues.
Comments on the Quality of English LanguageThe manuscript contained several grammatical errors and typos. Ensure thorough revision to correct these issues.
